# Dietary α-Linolenic Acid Counters Cardioprotective Dysfunction in Diabetic Mice: Unconventional PUFA Protection

**DOI:** 10.3390/nu12092679

**Published:** 2020-09-02

**Authors:** Jake S. Russell, Tia A. Griffith, Saba Naghipour, Jelena Vider, Eugene F. Du Toit, Hemal H. Patel, Jason N. Peart, John P. Headrick

**Affiliations:** 1School of Medical Science, Griffith University Gold Coast, Southport QLD 4217, Australia; jake.russell@griffithuni.edu.au (J.S.R.); tia.griffith@griffithuni.edu.au (T.A.G.); saba.naghipour@griffithuni.edu.au (S.N.); j.vider@griffith.edu.au (J.V.); j.dutoit@griffith.edu.au (E.F.D.T.); j.peart@griffith.edu.au (J.N.P.); 2VA San Diego Healthcare System and Department of Anesthesiology, University of California, San Diego, CA 92093, USA; hepatel@ucsd.edu

**Keywords:** α-linolenic acid, n-3 PUFA, diabetes, cardioprotection, caveolae, caveolins, cavins, inflammation, ischemia-reperfusion, mitochondria

## Abstract

Whether dietary omega-3 (n-3) polyunsaturated fatty acid (PUFA) confers cardiac benefit in cardiometabolic disorders is unclear. We test whether dietary α-linolenic acid (ALA) enhances myocardial resistance to ischemia-reperfusion (I-R) and responses to ischemic preconditioning (IPC) in type 2 diabetes (T2D); and involvement of conventional PUFA-dependent mechanisms (caveolins/cavins, kinase signaling, mitochondrial function, and inflammation). Eight-week male C57Bl/6 mice received streptozotocin (75 mg/kg) and 21 weeks high-fat/high-carbohydrate feeding. Half received ALA over six weeks. Responses to I-R/IPC were assessed in perfused hearts. Localization and expression of caveolins/cavins, protein kinase B (AKT), and glycogen synthase kinase-3β (GSK3β); mitochondrial function; and inflammatory mediators were assessed. ALA reduced circulating leptin, without affecting body weight, glycemic dysfunction, or cholesterol. While I-R tolerance was unaltered, paradoxical injury with IPC was reversed to cardioprotection with ALA. However, post-ischemic apoptosis (nucleosome content) appeared unchanged. Benefit was not associated with shifts in localization or expression of caveolins/cavins, p-AKT, p-GSK3β, or mitochondrial function. Despite mixed inflammatory mediator changes, tumor necrosis factor-a (TNF-a) was markedly reduced. Data collectively reveal a novel impact of ALA on cardioprotective dysfunction in T2D mice, unrelated to caveolins/cavins, mitochondrial, or stress kinase modulation. Although evidence suggests inflammatory involvement, the basis of this “un-conventional” protection remains to be identified.

## 1. Introduction

Dietary n-3 polyunsaturated fatty acids (PUFAs) may confer benefit in metabolic disease [1,2]; however, effects on homeostasis and disease risk are mixed [3,4]. There is support for protection against cardiovascular disease, including reduced disease risk together with benefit in those with existing ischemic heart disease (reductions in arrhythmias, hypertrophy, and coronary ischemia) [5]. There is also some evidence for improved myocardial resistance to ischemic insult, although such cardioprotection appears prominent in otherwise healthy hearts [6,7,8,9,10,11,12,13], and few studies assess outcomes in common comorbid conditions such as diabetes [14]. 

In humans, two essential fatty acids cannot be synthesized and must be acquired through the diet—a-linolenic acid (18:3n-3) and linoleic acid (18:2n-6) [2]. These are used to synthesize a variety of other unsaturated fatty acids, including the long chain PUFAs eicosapentaenoic acid (EPA) and docosahexaenoic acid (DHA) from α-linolenic acid (ALA), and arachidonic acid, from linoleic acid [2]. These may in turn be converted to eicosanoids and prostaglandins, which play important roles in inflammation and inflammatory disorders such as cardiovascular disease [2]. Due to competition between enzymes in each pathway, relative consumption of ALA and LA influences whether anti- or pro-inflammatory mediators will be synthesized [2]. While modulation of inflammation and associated pathways has been implicated in the biological effects of n-3 PUFAs, the mechanistic basis of benefit remains to be detailed. 

Modification and remodeling of membrane lipids and raft microdomains may broadly underlie shifts in survival and inflammatory signaling, gene expression, and metabolic and ionic homeostasis [1,15,16,17,18,19,20,21], including beneficial effects on gap junctions and connexin-43 [22,23,24]. Mitochondrial membranes may additionally be modified [25], potentially influencing energy and reactive oxygen species (ROS) generation. Nonetheless, how n-3 PUFAs specifically modify the structure and functionality of membrane domains awaits clarification [26]. Recent work indicates specific n-3 PUFAs increase cholesterol accumulation and molecular order, and raft or caveolar formation and size [27,28]. How the caveolin and cavin proteins governing caveolae formation, structure and function are modified is unclear, and no data are available regarding n-3 PUFA effects on these proteins in chronic diabetes. 

Dietary modulation of caveolar domains and proteins [29,30,31,32,33], mitochondrial function [22] or inflammation [34] might be of particular benefit in diabetes. Hearts of diabetic patients appear sensitized to damage [35] and desensitized to cardioprotection [36], and disruption of caveolae and caveolar proteins [29,30,31,32,37] governing ischemia-reperfusion (I-R) tolerance and cardioprotection [38,39,40,41,42] is implicated [42]. Similar mechanisms may underlie influences of aging on stress-resistance and survival signaling [43]. Cardiac caveolin expression and caveolar function appear repressed by both hyperglycemia [29,30,31] and saturated fats [32,33]. Although effects of T2D are less clear, we observe impaired cardiac I-R tolerance and ischemic preconditioning (IPC) in T2D mice [44], and pilot data support associated reductions in caveolin-3 expression [45]. Conversely, dietary PUFAs may improve myocardial caveolin-3 and survival signaling [46] and I-R tolerance [12,47,48,49] in healthy animals, together with metabolic homeostasis in insulin-resistant hearts [50], and caveolae-dependent cardioprotection in stressed hearts [6]. However, while dietary n-3 PUFAs reduce cardiac polysaccharide accumulation [51] and contractile and coronary dysfunction [52] in models of hyperglycemia or type 1 diabetes (T1D), effects on I-R tolerance and cardioprotection in chronic T1D or T2D are unknown. There are also limited data regarding other myocardial outcomes in T2D, though protection of sarcolemmal and mitochondrial membranes has been reported [22]. A single study also reports ALA-dependent improvements in post-ischemic outcomes and protein kinase B (p-AKT) in a model of acute T2D [53]; however, caveolae or caveolar proteins were not assessed, and opposing effects of acute vs. chronic diabetes on ischemic tolerance [53] render acute models of limited relevance to chronic disease [42]. No study has determined the effects of n-3 PUFA supplementation on myocardial I-R tolerance, cardioprotection or caveolar proteins in chronic T2D. 

We test the hypothesis that dietary ALA supplementation may be cardioprotective in T2D—including enhanced ischemic tolerance, survival signaling or responses to IPC [54,55,56,57,58,59]—via shifts in conventional PUFA-dependent mechanisms (caveolar makeup, kinase signaling, mitochondrial function, and inflammation) [46,60,61,62]. We employ a non-genetic model of chronic T2D development (β-cell stress + 21 weeks high-fat/high carbohydrate (HFHC) feeding) in young adult mice [44,63,64,65,66]. As discussed recently [44], this model is relevant to the pattern of T2D onset in humans, and avoids complications of adaptive cardioprotection evident in acute disease models, differing cardiac stress responses in early onset (e.g., genetic or disease-prone inbred models) vs. adult onset disease, and potential intrauterine programming of stress phenotype in genetic models. We focus on myocardial responses to I-R and IPC, while assessing basal expression of caveolins 1 and 3 (known to influence ischemic tolerance and IPC [39,40,67]) together with cavins 1 and 4: more recently identified determinants of caveolar structure/function and stress-resistance [68,69,70] whose sensitivities to dietary lipids are unknown.

## 2. Materials and Methods

### 2.1. Experimental Animals and Ethics

All investigations were approved in accordance with the policy guidelines (The Animal Care and Protection Act 2001) of the Animal Ethics Committee of Griffith University (ethics approval MSC/14/16/AEC), which is accredited by the Queensland Government, Australia. Male C57Bl/6 mice were supplied by the Animal Resource Centre (Perth, Australia) and housed in the Griffith University Animal Facility for the duration of the study. Mice were habituated to the facility for at least 1 week prior to studies and were housed in groups of 4, with sawdust bedding and ad lib access to water and food. The mice were maintained in a 12-h day/night lighting cycle at a constant temperature of 21 °C and 40% humidity. An outline of the experimental design, including details of analytical procedures and “*n*” values, is provided in Appendix A.

### 2.2. Murine T2D Model and ALA Supplementation

A non-genetic model of chronic adult-onset T2D was used, as detailed in our recent work [44], and employed by others [63,64,65,66]. Male C57Bl/6 mice (8 week old) were fasted and administered a single intraperitoneal injection of streptozotocin (STZ, 75 mg/kg) to stress but not eliminate pancreatic b-cells, and switched from standard rodent chow to a Western HFHC diet for 15 weeks (43.2% kcal as fat; 39.7% carbohydrates; 17.1% protein; sourced from Specialty Feeds, WA, Australia, and equivalent to Teklad diet TD08811). After 15 weeks, half were randomly switched to ALA supplementation for the final 6 weeks, while half continued HFHC feeding alone (*n* = 22/group). We selected ALA based on evidence this n-3 PUFA can modify myocardial caveolin-3 expression [42] and may confer cardiac benefits in diabetes [53]. The ALA supplemented diet contained 10% of total fat from ALA, as previously described [71], and was calorie- and macronutrient-matched to the HFHC diet. A subset of age-matched non-diabetic (control) mice (*n* = 8) were included to confirm that the combination of STZ/HFHC feeding induces a T2D phenotype. After the 21 week experimental protocol, mice were sacrificed and hearts immediately removed for either analysis of myocardial caveolar protein and kinase expression or mitochondrial function in normoxic tissue (*n* = 6/group), or Langendorff perfusion to assess contractile function, intrinsic I-R tolerance (*n* = 8/group) and responses to IPC (*n* = 7–8/group).

### 2.3. Assessment of Metabolic Phenotype

Body weights (Figure 1) and fasting glucose were measured in all non-diabetic (CTRL, *n* = 8) and diabetic mice (T2D, *n* = 22; T2D + ALA, *n* = 22) (Appendix A). Glucose tolerance tests (GTTs) and fasting insulin measurements were also undertaken in all control mice. To obtain representative phenotype data for the T2D groups (while constraining costs), GTTs were undertaken in 12 randomly selected mice/group, and fasting insulin (and associated glucose) was assayed in 10 mice/group (Appendix A). These “*n*” values provide more than sufficient power to detect predicted metabolic changes, based on our prior work in this model [44]. However, one T2D mouse was ultimately identified as an outlier (based on abnormally high insulin) via Grubbs test, and was removed from all analyses (see Statistical Analyses).

The GTTs were performed at 14 (1 week pre-ALA) and 20 weeks (5 weeks post-ALA) to assess emergence of the diabetic phenotype. Mice were fasted for 4 h before baseline blood glucose was measured via tail prick (Accu-Check II glucometer; Roche Diagnostics, Castle Hill, Australia). Mice were then administered a 20% glucose solution (2 g glucose/kg, IP injection) with glucose levels monitored every 30 min for 3 h. The area under the curve (AUC) was calculated from individual datasets to yield a single index of glucose clearance.

Fasting glucose was measured via tail snip at 14 (1 week pre-ALA) and 20 weeks (5 weeks post-ALA) in all mice, with these data provided in Appendix B. Fasting insulin (and glucose) and insulin sensitivity were additionally measured at 14 and 20 weeks in a sub-set of T2D and T2D + ALA mice. This fasted blood was immediately assayed for glucose, or collected into ethylenediaminetetraacetic acid (EDTA) coated tubes, and incubated for 30 min at room temperature before 5 min centrifugation at 10,000 g. Serum was collected and stored at −80 °C until insulin analysis via enzyme-linked immunosorbent assay (ELISA) according to manufacturer instructions (Crystal Chem Inc., Elk Grove Village, IL, USA). Fasting insulin and glucose were used to calculate two measures of insulin sensitivity, the homeostatic model assessment of insulin-resistance (HOMA-IR) and quantitative insulin sensitivity check index (QUICKI):

HOMA-IR = fasting insulin (ng/mL) × fasting blood glucose (mg/dL)/405.

QUICKI = 1/ (log[fasting insulin μU/mL] + log[fasting glucose mg/dL]).

Both correlate with direct measures of insulin sensitivity in mice [72] and rats [73]. Sufficient serum sample remained to additionally assay circulating leptin in 8 mice/group and total cholesterol in 7 mice/group), using an ELISA (CSB-E04650m-96T; Cusabio, College Park, MD, USA) and colorimetric/fluorescent assay (#ab65390; Abcam, Melbourne, VIC, Australia), respectively.

### 2.4. Heart Perfusion and Responses to I-R ± IPC

Hearts were Langendorff perfused for assessment of baseline (normoxic) function, I-R tolerance, and efficacy of IPC [74]. Briefly, mice were anesthetized with sodium pentobarbital (60 mg/kg, intraperitoneal injection), hearts excised, and the aorta immediately cannulated for perfusion of the coronary circulation with modified Krebs–Henseleit buffer, gassed with 95% O_2_/5% CO_2_, maintained at 37 °C (pH 7.4) and containing: 119 mM NaCl, 11 mM glucose, 22 mM NaHCO_3_, 4.7 mM KCl, 1.2 mM MgCl_2_, 1.2 mM KH_2_PO_4_, 1.2 mM EDTA, and 0.5 mM and 2.5 mM CaCl_2_.

Contractile function was monitored via fluid-filled balloon in the left ventricle, inflated to an end-diastolic pressure (EDP) of 5 mmHg [74]. Coronary flow was measured via ultrasonic flow-probe proximal to the aortic cannula and connected to a T206 flowmeter (Transonic Systems Inc., Ithaca, NY, USA). A 4 channel MacLab system (ADInstruments Pty Ltd., Castle Hill, Australia) connected to an Apple iMac computer was used for continuous acquisition (1 KHz sampling rate) and processing of data, including: left ventricular EDP, systolic, and developed pressures(LVDP), +dP/dt and −dP/dt (positive and negative differentials of pressure change over time), heart rate, and coronary flow. Perfusate temperature was continuously monitored via thermal probe connected to a Physitemp TH-8 digital thermometer (Physitemp Instruments Inc., Clifton, NJ, USA).

Hearts were stabilized at intrinsic beating rates for 20 min before ventricular pacing at 420 beats/min (via silver wires attached to SD9 stimulator; Grass Instruments, Quincy, MA, USA). After 10 min, hearts were subjected to 25 min normothermic global ischemia followed by 45 min aerobic reperfusion (*n* = 8/group). For IPC, a subset of hearts was subjected to 3 × 5 min episodes of ischemia separated by 5 min reperfusion prior to the index ischemia (*n* = 7 for T2D and 8 for T2D + ALA). This is a substantially greater stimulus than the 3 × 1.5 min algorithm we previously showed is effective in hearts of healthy mice [43] and has been shown to overcome inhibitory effects of diabetes in rat hearts [75]. To assess myocardial disruption and cell death, total post-ischemic efflux of lactate dehydrogenase (LDH) was measured in each heart [8,43,68,76,77]. We confirm this common measure of cell death is strongly correlated with infarct size in this murine model [76]. Total coronary effluent throughout 45 min of reperfusion was collected on ice and stored at −80 °C until analysis using a CytoTox 96^®^ NonRadioactive Cytotoxicity Assay (G1780, Promega; Madison, WI, USA), per manufacturer’s instructions. Total LDH efflux was normalized to heart weight and expressed relative to the untreated T2D group. To assess apoptosis, nucleosome content in post-ischemic myocardial lysate was assayed via ELISA, per manufacturer’s instructions (Roche Cell Death Plus, 11774425001). Cardiac nucleosome contents were also normalized to levels in the untreated T2D group.

### 2.5. Myocardial Tissue Fractionation and Protein Analysis

Left ventricular myocardium was divided into two samples which were snap frozen in liquid N_2_ and stored at −80 °C until analysis (*n* = 6/group). To enrich for cytosolic elements, ventricular tissue was homogenized in ice-cold lysis buffer containing protease inhibitors: 1 mM PMSF, 10 µM leupeptin, 3 mM benzamidine, 5 µM pepstatin A, and 1 mM NaO. To enrich for mitochondria, ventricular samples were homogenized in ice-cold isolation buffer, comprised of the aforementioned lysis buffer with addition of 70 mM sucrose, 190 mM mannitol, 20 mM HEPES, and 0.2 mM EDTA. The homogenate was centrifuged at 600 g for 10 min at 4 °C, with the supernatant (mitochondria, cytosol and plasma membrane) re-centrifuged at 100,000 g for 1.5 h (to obtain cytosolic fraction) or 10,000 g for 30 min (to obtain mitochondrial fraction) at 4 °C. The supernatant enriched for cytosol was transferred to a new tube and stored at −80 °C. The mitochondrial pellet was washed in isolation buffer at 600 g for 10 min before being re-suspended in lysis buffer and storage at −80 °C. Protein content was determined via a Pierce™ BCA Protein assay (Thermo Fisher Scientific; Scoresby, Australia), and aliquots containing 30 μg protein were prepared. Expression of total and phosphorylated AKT and glycogen synthase kinase-3β (GSK3β) was measured in the cytosolic fraction, while caveolin-1 and -3 and cavin-1 and -4 were measured in whole cell lysate and mitochondrial fractions, with Western immunoblot data normalized to actin expression.

Membrane sub-fractionation: Sucrose density fractionation was employed to obtain caveolae-enriched fractions from ventricular samples (*n* = 6/group). Left ventricles were homogenized in a glass dounce on ice with pH 11 lysis buffer containing: 150 mM Na_2_CO_3_, 1mM EDTA, 1 mM PMSF, 10 µM leupeptin, 3 mM benzamidine, 5 µM pepstatin A, and 1 mM NaO. Samples were then briefly sonicated (3 × 10–15 s cycles with 1 min intervals, on ice) and incubated on ice for 10 min. Protein content was determined via Pierce™ BCA Protein assay (Thermo Fisher Scientific; Scoresby, Australia), and 850 µL of normalized whole cell lysate containing 5 mg protein was loaded into an ultracentrifuge tube. A solution containing 25 mM MES, 150 mM NaCl, and 2 mM EDTA was used to generate sucrose solutions containing 80%, 35%, and 5% sucrose (*w*/*v*). Homogenate in ultracentrifuge tubes was then mixed with 850 µL 80% sucrose to generate 1.7 mL 40% sucrose. Above the 40% layer, 5.1 mL of 35%, followed by 3.4 mL of 5% sucrose were carefully layered. The mixture was centrifuged at 175,000 g using a Beckman Coulter SW41 Ti rotor for 18 h at 4 °C. Sequential fractions (1–12) were obtained by carefully removing 850 µL aliquots from the top-down, and stored in −80 °C. Western blot analysis confirms caveolin-3 abundance in buoyant fractions 4–6, indicative of caveolae-enriched compartments, as previously described [78,79]. Equal volumes of fractions 4–6 were pooled and referred to as the buoyant fraction. Western blots were performed to assess expression of caveolin-1, caveolin-3, cavin-1, and cavin-4 in equal volumes of buoyant fraction, with data normalized to β-actin expression.

Immunoblot analysis: Once thawed, protein aliquots were prepared with equal volumes of loading dye and denatured at 95 °C for 5 min in a heating block. A 30 µL volume of each sample was loaded into hand-cast 10% acrylamide gels. Protein separation was achieved by running gels at 150 V for 60 min. Transfer of proteins was achieved using a polyvinylidene difluoride membrane at a constant 75 V for 1.5–2 h, with blocking with Odyssey fish serum for an additional 2 h at room temperature. Transferred proteins were incubated with primary antibodies overnight at 4 °C with gentle agitation: caveolin-1 (1:500; rabbit polyclonal, ab2910, Abcam); caveolin-3 (1:500; rabbit polyclonal, ab2912, Abcam); cavin-1 (1:500; rabbit polyclonal, ab48824, Abcam); cavin-4 (1:500; rabbit polyclonal, 55464-1-AP, Proteintech); p-AKT (1:1000; rabbit polyclonal, 9271, Cell Signaling Technology, Inc., Danvers, MA, USA); total AKT (1:1000; rabbit polyclonal, 9272S, Cell Signaling Technology, Inc.); p-GSK3β (1:1000; rabbit polyclonal, 9336, Cell Signaling Technology, Inc.); total GSK3β (1:1000; rabbit monoclonal, 9315, Cell Signaling Technology, Inc.); glyceraldehyde 3-phosphate dehydrogenase (GAPDH; 1:1000; mouse monoclonal, sc-32233, Santa Cruz Biotechnology, Dallas, TX, USA); or β-actin (1:1000; mouse monoclonal, sc-8432, Santa Cruz Biotechnology).

The membrane was washed 4 times in tris-buffered saline and Tween 20 (TBST) and again in tris-buffered saline (TBS; 5 min each) before incubation with corresponding secondary antibodies at room temperature in the dark: IRDye^®^ 680RD donkey anti-mouse 1:30,000 (925-68072, LI-COR); or IRDye^®^ 680RD goat anti-rabbit 1:30,000 (925-68071, LI-COR). Membranes were then re-washed 4 times in TBST and once in TBS (5 min each), before drying overnight between paper towels in the dark. Membranes were visualized on a Li-Cor Odyssey Infrared Imaging System (Li-Cor Biosciences, Lincoln, NE, USA) and protein densitometry data for each sample normalized to the actin (found to be consistent among all groups).

### 2.6. Mitochondrial Respiratory Function

Respiratory function in fresh right ventricular tissue (*n* = 6/group) was assessed using an Oxygraph-2k instrument (Oroboros Instruments, Innsbruck, Austria), at 37 μC with a substrate-uncoupler-inhibitor-titration (SUIT) protocol. Briefly, 8–10 mg left ventricular myocardium was shredded with a PBI-shredder HRR-Set (Oroboros Instruments, Innsbruck, Austria) and added to oxygraph chambers (2 mg/mL tissue per chamber) stabilized with Mir05 media and catalase (280 U/mL). Following baseline calibration (5 min), pyruvate (5 mM), malate (2 mM), and glutamate (10 mM) were added prior to ADP (2–4 mM) to assess complex I leak and oxidative phosphorylation linked respiration respectively. Cytochrome *c* (10 μM) was added to measure mitochondrial membrane integrity, followed by succinate (10 mM) to determine Complex I & II linked respiration. Carbonyl cyanide m-chlorophenyl hydrazone (FCCP, 0.5 μM) was used to measure maximum respiratory capacity. Complex I and III inhibition with rotenone (1 μM) and antimycin A (5 mM) respectively, provided measurement of residual oxygen consumption (ROX).

### 2.7. Myocardial and Hepatic Inflammatory Mediator Profiles

To explore potential shifts in inflammatory signaling, a Mouse XL Cytokine Array Kit (ARY028, R&D Systems, Minneapolis, MN, USA) was used to compare expression of 111 proteins between pooled cardiac lysates samples (*n* = 6 per group) from untreated and ALA treated lysates, according to the manufacturer instructions. After subtraction of the optical density for negative control spots, the density for each pair of protein spots was normalized to the mean density of reference spots on each array. A minimum threshold for inclusion of 2× the zero blank density was employed, and only proteins exhibiting a ≥1.5 fold change up or down (T2D + ALA vs. T2D) were highlighted.

PCR analysis: Cardiac transcripts for key inflammatory mediators linked to detrimental influences of obesity, saturated fats, and T2D on insulin signaling, ER stress, autophagy and cell death were quantitated via PCR in sub-sets of hearts from T2D ± ALA mice (*n* = 6/group). Primer details for *Nfkb1*, *Rela*, *Tnf*, *Il1b*, *Hmgb1*, and *Tlr4* are provided in Appendix C. Total RNA extraction from ventricular and hepatic tissue samples was performed using the Maxwell^®^ RSC simplyRNA Tissue Kit (Promega Corporation, Alexandria, NSW, Australia) on the Maxwell^®^ RSC instrument (Promega Corporation, Alexandria, NSW, Australia), per manufacturer’s instructions. RNA was eluted into nuclease-free water, and RNA concentration obtained using the *QuantiFluor*^®^
*RNA* System (Promega Corporation, Alexandria, NSW, Australia) prior to storage at −80 °C. Total RNA in each sample was adjusted to 150 ng/uL prior to cDNA synthesis with the RevertAid First Strand cDNA Synthesis Kit (Thermo Fisher Scientific, Scoresby, VIC, Australia), according to instructions for Oligo (dT)18 Primer preparations. Prepared cDNA was stored at −20 °C until PCR analysis.

The qPCR reaction contained 5 µL 2 × SYBR green Mastermix (Bio-Rad), 1 µL of each primer (300–500 nM) and 1 µL of DNA template in a 10 µL reaction volume. PCR analysis was performed using an Applied Biosystems StepOnePlus thermocycler, with the following cycle program: 10 min at 95 °C, followed by 40 cycles of 15 s at 95 °C, and 1 min at 60 °C. Specificity of amplification was checked by melting curve analysis, which was carried out by a final melting of 15 s at 95 °C followed by dissociation curve construction comprising increasing temperature by 0.3 °C increments from 60 °C. Sterile water, RNA samples without addition of reverse transcriptase in cDNA synthesis, and internal control cDNA samples were used as controls. The linearity of each qPCR target was tested by preparing a series of dilutions of the same cDNA stock to determine PCR efficiency. The target gene was normalized to the housekeeping gene Phosphoglycerate kinase 1 (*Pgk1*). The housekeeping gene was run on every plate for each sample. Each plate contained duplicate samples for the target gene and the housekeeping gene. Analysis of PCR data was conducted using the 2^−ΔΔCT^ method [80].

### 2.8. Statistical Analyses

Data were analyzed using GraphPad Prism 8, and are presented as means ± SEM. Specific a priori hypotheses included (i) IPC is ineffective in T2D hearts; (ii) ALA improves intrinsic I-R tolerance; (iii) ALA improves IPC efficacy; and (iv) ALA improves expression or phosphorylation of caveolar proteins, AKT and GSK3β and reduces inflammatory markers. A Grubbs test was employed to identify outliers: one mouse in the T2D group was removed from all analyses, due to outlying insulin levels. An ANOVA with planned comparisons was employed to specifically test these hypotheses, eliminating nonsensical contrasts constraining statistical power [81]. Fisher’s LSD post-hoc test was employed for specific comparisons, while a Student’s t-test was used for comparisons between two groups. A *p*-value < 0.05 was considered indicative of statistical significance across tests.

## 3. Results

### 3.1. Metabolic Phenotype

Consistent with metabolic outcomes reported in similar T2D models [44,63,64,66,82,83], mice exhibited high body weights (Figure 1A), moderate fasting hyperglycemia (Figure 1B) and glucose intolerance (Figure 1B,C). Although trends to increased fasting insulin levels (Figure 1D) in T2D mice did not reach statistical significance compared to controls prior to ALA supplementation (*p* = <0.10), the HOMA and QUICKI calculations indicate mice were insulin-resistant (Figure 1E) and had impaired insulin sensitivity (Figure 1F), respectively. Altogether, outcomes match those previously reported in this model [44,63,64,66,82,83], and compare with lower body weights and glucose and insulin, levels, and higher insulin sensitivity in age-matched control mice. Subsequent supplementation with ALA did not significantly alter body weight, glucose, or insulin handling compared to non-supplemented T2D mice. Although food intake was not measured, comparable body weights with calorie-matched diets suggest similar food intakes. We then specifically tested whether ALA consumption alters circulating leptin or cholesterol levels in T2D mice. Circulating leptin was reduced by ALA supplementation (Figure 1G), whereas cholesterol levels were unchanged (Figure 1H).

### 3.2. Myocardial Function, I-R Tolerance and Responses to IPC

Baseline contractile function and coronary flow were not significantly modified by ALA supplementation (Appendix D). Myocardial contracture during index ischemia was also unaltered with ALA supplementation, while IPC accelerated contracture in both groups of hearts (Figure 2). Acceleration of contracture is a known effect of IPC stimuli [84] and may potentially limit cardiac benefit with this stimulus. No additional changes in contracture development were observed with a combination of IPC in hearts from ALA mice.

Neither IPC or ALA independently altered final post-ischemic recovery of LVDP, total LVDP area under the curve (overall contractile recovery throughout reperfusion) (Figure 3A), or recovery of EDP (Figure 3B). However, ALA supplementation resulted in functional protection with IPC, which significantly reduced contractile and diastolic dysfunction in ALA treated (but not untreated) T2D hearts. Increased post-ischemic LDH efflux (Figure 3C) indicates IPC worsened myocardial death in T2D hearts, a paradoxical injury that was eliminated by ALA supplementation. In contrast, a marker of post-ischemic apoptosis (nucleosome content) was not altered with either ALA or IPC (Figure 3D).

### 3.3. Cardiac Caveolar Proteins and Survival Kinases

Supplementation with ALA had no significant effect on total, buoyant membrane, or mitochondrial fraction expression of caveolin-1, caveolin-3, cavin-1, and cavin-4 in ventricular tissue (Figure 4). Baseline expression and phosphorylation of cytosolic AKT and GSK3β was also unchanged with ALA supplementation (Figure 5).

### 3.4. Cardiac Mitochondrial Function

Mitochondrial respiratory functional parameters are presented in Figure 6. Data reveal no significant effects of ALA on Complex I or II O_2_ flux (Figure 6A,B); leak O_2_ flux (Figure 6C); oxidative phosphorylation (OxPhos) capacity O_2_ flux (Figure 6D,E); electron transfer system (ETS) capacity O_2_ flux (Figure 6F); or flux control ratios (Figure 6G,H).

### 3.5. Myocardial Inflammatory Mediators

Protein arrays revealed modest and mixed effects of ALA on inflammatory modulator expression (Figure 7). Protein expression was only modified by up to ~2.5-fold with ALA, including evidence of up-regulation of VEGF, MCP-1, M-CSF, VCAM-1, ICAM-1, CD40, and interleukins 7, 13, 15, and 33, while WISP-1 was repressed. Since expression of several proteins of interest was close to background, we additionally assessed transcript levels for select mediators (Figure 8). Broadly consistent with proteome outcomes, analysis of gene expression revealed no changes in cardiac expression of *Nfkb1*, *Rela*, *Il1b*, or *Tlr4* transcripts; however, there was a marked reduction in *Tnf* together with an insignificant trend to reduced *Hmgb1* expression with ALA supplementation (Figure 8).

## 4. Discussion

Our primary goal was to test whether PUFA supplementation confers myocardial benefits in a model of T2D, and whether this was associated with conventional PUFA-dependent mechanisms. Dietary PUFAs may be cardioprotective [6,46,53,85], including evidence of improved myocardial phenotype and function in diabetes [51,52,86,87,88]; however, controversy remains regarding mechanisms and impacts of dietary PUFAs in both health and disease. We find that six weeks ALA supplementation was of selective benefit in hearts of T2D mice, improving cardiac IPC without influencing intrinsic I-R tolerance. This was associated with mixed changes in inflammatory mediators (including reduced *Tnf*). However, there were no changes in expression or sub-cellular localization of caveolin and cavin proteins, despite reported effects of n-3 PUFAs on raft microdomain formation, size, and structure [26,27,28]. Moreover, benefit with ALA was not associated with changes in metabolic phenotype, baseline mitochondrial function, or baseline expression of p-AKT or p-GSK3β.

### 4.1. Systemic Phenotype

Data confirm metabolic compromise and a T2D-like phenotype in the present model, including elevated glucose and insulin levels, and impaired glucose clearance and insulin sensitivity (Figure 1). The outcomes are consistent with our prior findings [44] and other studies employing this model [63,64,66,82,83]. These metabolic factors are well-established risks for ischemic heart disease [89] and were unaltered by ALA supplementation (Figure 1). This agrees with meta-analyses indicating no effects of n-3, n-6 or total PUFA intake on T2D risk or outcomes [4,90]; although, impacts in diabetes remain contentious [34,90,91,92,93]. On the other hand, multiple trials show substitution of saturated fatty acids with PUFAs can lower blood glucose, HbA1c, and HOMA-IR in healthy adults [94]. Metabolic effects thus differ in health vs. disease and may also be PUFA specific: insulin-sensitivity is related to adipose ALA but not EPA or DHA levels in healthy adults [95], and marine n-3 PUFAs induce distinct protective effects compared with ALA [96]. Additionally, recent meta-analysis indicates EPA and DHA do not influence cardiovascular disease risk, whereas there is evidence (albeit low quality) for protection with ALA [3].

Studies of PUFA supplementation in models of chronic diabetes are limited. There is evidence ALA enriched (Chia seed) [34] or high polyunsaturated:monounsaturated fatty acid ratio diets [92] improve insulin signaling and glucose handling in models of obesity or T1D. Other studies support n-3 PUFA dependent improvements in glucose homeostasis, dyslipidemia, and body weight in rodent models of T1D [51,97], although the latter weight change is not always observed [98]. Consistent with our findings, dietary n-3 PUFA does not modify hyperglycemia or dyslipidemia in mouse and rat models of T2D [99,100]. On the other hand, n-3 PUFA reportedly reduces hyperglycemia and dyslipidemia in the Goto–Kakizaki rat model [22], and dietary ALA improves glycemic control in *ob^−^/ob^−^* mice [101]. Others report that PUFA supplementation may exaggerate hyperglycemia and dyslipidemia in a rat model of T2D [102]. The basis for these varying (sometimes opposing) outcomes is unclear; however, as we note above, there are important drawbacks to genetic and inbred models of disease development, particularly in terms of stress-resistance and survival signaling [22]. Differences in duration and level of dietary PUFA supplementation may also influence outcomes. While some studies (including here) assess 4–6 weeks of supplementation [97,98,100,101], others extend this to ~8 [22], 10 [51], or 16 weeks [99,102], which may increase biological outcomes. The degree of PUFA supplementation may also be relevant; we study a modest supplementation of 10% calorie content as ALA (matched by reduced saturated fat), while some studies employ much higher levels, for example, replacement of ~50% of dietary fat with PUFAs [99,102]. It is also relevant to note that increased ALA consumption may not necessarily promote DHA synthesis [103], suggesting that a mix of dietary n-3 PUFAs could be more effective in conferring cardiac benefits.

### 4.2. Myocardial Function, Ischemic Tolerance and Preconditioning

Baseline contractile function and coronary perfusion were insensitive to ALA supplementation, consistent with lack of mechano-energetic effects of PUFA supplementation in the hearts of healthy animals [104,105], despite improvements in mitochondrial function [106]. In contrast, others provide evidence n-3 PUFAs may improve function in healthy hearts [10] or promote cardiac efficiency without modifying contractility [13,107]. Contractile function and coronary flow in hearts of T1D rats are also reportedly improved with n-3 PUFA supplementation [86,87,88].

Protection against myocardial damage with infarction or surgical I-R remains a highly desirable goal that may prove particularly challenging in diabetes, which may sensitize the heart to injury while desensitizing it to protective stimuli [42]. This is the first evidence ALA intake improves myocardial preconditioning in chronic T2D, although intrinsic I-R tolerance was unaltered. Curiously, McLennan and colleagues observe the reverse in healthy hearts: n-3 (not n-6) PUFA supplementation enhanced ischemic tolerance without influencing IPC [11]. Others report no effect of n-3 PUFA supplementation on myocardial ischemic tolerance in healthy animals [108]; however, there is also evidence of improved I-R tolerance in non-diabetic animals [6,7], including reduced LDH efflux, and contractile or respiratory dysfunction [8,9,10,11,12,13]. Interestingly, direct infusion of n-3 PUFAs may also confer acute protection during reperfusion of otherwise healthy hearts [109], including parallel changes in infarction and markers assessed here (LDH efflux, contractile dysfunction, p-AKT, and p-GSK3β. Lack of effect on ischemic tolerance also contrasts cardioprotection with ALA in a model of acute T2D [53], associated with enhanced survival and suppressed inflammatory signaling. However, this model may not be relevant to chronic disease. Our data indicate that although PUFA supplementation counters the injurious impacts of IPC (and restores functional protection) in T2D hearts, other effects reported in healthy animals [12,47,48,49] or in acute disease [53] are not apparent in the context of chronic T2D. This may reflect a T2D-dependent increase in the “threshold” for cardioprotection [75], necessitating greater/additional stimuli to trigger protection. Tsang et al. report that while a single 5 min IPC episode protects hearts from healthy rats, three episodes are required to elicit protection in a transgenic model of T2D [75]. The T2D mice here were unresponsive to this 3 × 5 min IPC protocol, a stimulus substantially greater than previously shown by us to protect healthy hearts [43]. Indeed, IPC worsened myocardial damage in the hearts of T2D mice (Figure 3C).

### 4.3. Caveolar Protein Expression and Localization

Dietary n-3 PUFAs incorporate into plasma membranes to modify biophysical properties, and lipid raft/caveolae makeup and function [1,18,19,20,21,110]. Indeed, n-3 PUFAs may regulate the formation, size, and function of caveolae to influence diverse receptor and second messenger signaling paths [92], substrate uptake [111], ion fluxes and electrical coupling/stability [112]. Myocardial stress-resistance and cardioprotection are strongly dependent on these domains, and associated caveolins [113] and cavins [68]. Caveolin-3 and cavin-1 are essential to the structural integrity and function of caveolae [114], and to myocardial I-R tolerance and cardioprotection [39,40,68]. Caveolin-1 may also be protective [67], whereas cavin-4 appears to limit cardiac stress-resistance [70]. Prior studies indicate ALA may augment caveolin-3 in cardiomyocytes and cardiomyopathic hearts [46], and counters both inflammation-dependent down-regulation of skeletal myoblast caveolin-3 [60] and up-regulation of endothelial caveolin-1 [115]. Other data indicate n-3 PUFA intake suppresses vascular caveolin-1, albeit in a model of experimental menopause [62]. We find no evidence of ALA-dependent changes in the expression or sub-cellular localization of caveolins 1 and 3 in T2D hearts, or of cavin 1 or 4 expression, also critical to caveolar structure/function, cardiac stress-resistance and signaling [68,69,70]. Whether this reflects a limited capacity of dietary ALA to modify cardiac microdomains, compared with DHA or EPA for example, remains to be tested; there is evidence of differing membrane incorporation and cardiovascular influences of PUFA sub-types [3,27,28,95,96].

Since neither total, membrane microdomain, or mitochondria associated caveolin and cavin levels were modified with ALA, we find no evidence for PUFA-dependent changes in either caveolar levels or translocation to mitochondria, which has been linked to cardioprotection [116]. Although Folino et al. report caveolae-dependent protection with ALA in stressed cardiomyoblasts (while effects of caveolar disruption in un-stressed cells were untested) [6], this does not exclude effects on distal elements of caveolae-dependent signaling, including effector molecule expression/activation (e.g., survival kinases and eNOS), intracellular transduction to mitochondrial targets, and subsequent modification of associated death signaling [39,40,68]. While lack of change in sarcolemmal caveolins and cavins is inconsistent with an ALA-dependent increase in overall caveolar size or density, it remains possible caveolar structure and biophysical properties are modified, influencing interactions between caveolar proteins and signaling complexes. This might be assessed in future work, including interactions between caveolar proteins and membrane receptors and effectors (including NOS) [17,117].

### 4.4. Myocardial Kinase Expression and Phosphorylation

Lack of change in myocardial AKT and GSK3β expression or phosphorylation (Figure 5) is internally consistent with failure of ALA to modify caveolar proteins known to influence these kinases [118]. Contrasting these observations, Xie et al. found ALA feeding increased post-ischemic p-AKT in a model of acute T2D but not otherwise healthy hearts [53], while acute infusion of n-3 PUFA emulsion enhances post-ischemic p-AKT and p-GSK3β in hearts from healthy mice [109]. However, baseline expression/phosphorylation was not assessed in either of these studies, and the acute T2D model is not reflective of chronic disease outcomes [42]. Other findings support select effects of n-3 PUFA on AKT phosphorylation in stressed but not control myocardial [85] or other cell types [56]. Studies in non-cardiac cells indicate ALA/n-3 PUFA increases AKT phosphorylation in aged tissue [119] or cancer cells [54], and in settings of metabolic stress, including high fat feeding [57,59], and exposure to saturated fats [56] or hyperglycemia [57,59]. In contrast, n-3 PUFAs reportedly inhibit GSK3β phosphorylation in other cell types [120]. Lack of effect of ALA on baseline phosphorylation here suggests effects on AKT/GSK3β reported in healthy hearts or other disease models may be impaired in the context of chronic T2D.

### 4.5. Mitochondrial Function

Given the importance of mitochondria in responses to ischemic or other insult, we speculated that ALA supplementation protects T2D hearts via improved mitochondrial makeup/function [121]. Supplementation with n-3 PUFAs may protect mitochondria [122], and there is evidence cardioprotection with n-3 PUFAs involves modulation of mitochondrial lipids [12] and mitochondrial preservation [8]. We find no effect of ALA on baseline mitochondrial function, while mitochondrial responses to I-R or IPC were not tested. Effects of n-3 PUFAs on cardiac mitochondria are varied, including: no effect on baseline function vs. modest improvements in post-ischemic respiration [10]; inhibitory effects on complex I and IV activities [123]; or stimulatory effects on complexes I–V [8]. Others report that n-3 PUFAs modify makeup but not function in cardiac mitochondria from insulin-resistant obese animals [124]. Our data are consistent with some of these observations [10] and leave open the possibility that ALA could selectively modify mitochondrial function in post-ischemic but not un-stressed myocardium [10].

### 4.6. Inflammatory Mediators

Inflammatory signaling plays an important role in the cardiac abnormalities in T2D, contributing to insulin-resistance, ER stress, and shifts in autophagy and apoptosis [125,126,127]. While PUFA supplementation is attributed with broad anti-inflammatory effects [128], our data reveal very select influences of ALA in T2D hearts, including a marked repression of *Tnf* transcript and a trend to reduced *Hmgb1*, without changes in NF-κB related transcripts, *Il1b* or *Tlr4*. Reductions in encoded TNF-α or HMGB1 are predicted to confer protection. Exploratory proteomic analysis also revealed mixed effects of ALA, including unexpected pro-inflammatory and adhesion molecule responses that may be detrimental. However, improved expression of Vascular endothelial growth factor (VEGF) may be of benefit, linked to cardioprotection with pre- and post-conditioning stimuli [129,130].

The basis of this selective benefit with ALA is unclear, arising independently of systemic metabolic phenotype, expression/localization of caveolins and cavins, and baseline mitochondrial function, and AKT and GSK3β expression/phosphorylation. We recently found that post-ischemic AKT levels and phosphorylation are unaltered by IPC or T2D [44], leaving the possibility that phosphorylation patterns during the IPC stimulus and/or initial minutes of reperfusion might be modified with ALA; AKT phosphorylation at these times is linked to I-R outcomes and cardioprotection. Shifts in select inflammatory mediators and VEGF could participate in improved IPC with ALA, while excess TNF-α and HMGB1 are both linked to myocardial injury; TNF-α signaling is also implicated in cardiac preconditioning [131] and HMGB1 may additionally precondition hearts via up-regulating protective VEGF [132], which was augmented by ALA here. Finally, it is also notable that n-3 PUFAs are highly pleiotropic [133], and could influence IPC through cytochrome P450 (e.g., epoxyeicosanoid formation) and arachidonate metabolic pathways [134], inter-related endocannabinoid signaling [135], activation of key transcriptional regulators [136], and/or inhibition of oxidative stress [137].

### 4.7. Limitations and Future Studies

Several limitations are worth mentioning. First, since our hypothesis that PUFA supplementation protects the heart through altered expression/localization of caveolar proteins was not supported, other fundamental mechanisms participate and remain to be identified. Evidence here suggests a potential role for altered inflammatory and VEGF signaling, while modification of the biophysical properties of caveolae independent of caveolar protein levels remains a possibility that warrants investigation. Shifts in caveolar associated proteins known to influence I-R tolerance may also be relevant, for example the gap junction protein connexin-43 is linked to caveolae and its expression and/or phosphorylation state may be modified with diabetes [138].

An added limitation is that we employ a surrogate measure of myocardial disruption/death (total LDH loss), and more specific interrogation of select death processes might be of value. This measure correlates strongly with infarct size in the model used here [43], and reflects death involving loss of membrane integrity (e.g., oncosis). Since nucleosome levels—a marker of apoptosis—were unaltered, our data support select PUFA effects on IPC dependent oncosis and functional protection, independently of apoptosis or baseline mitochondrial function, caveolar protein levels or survival kinase expression. Future studies could also test whether ALA supplementation prior to or concurrent with disease development might delay or prevent cardiometabolic dysfunction and disease, testing value as a preventative measure. That said, meta-analyses indicate n-3, n-6, or total PUFA intake has no influence on T2D risk or outcomes [4,90], despite evidence PUFAs can improve glucose levels and insulin sensitivity in healthy adults [94].

## 5. Conclusions

The present study indicates that six weeks of moderate dietary ALA supplementation (10% of dietary fat calories) does not improve myocardial I-R tolerance in chronic T2D, yet counters detrimental effects of IPC and restores functional protection. This selective benefit arises in the absence of changes in systemic metabolic phenotype, myocardial expression, and sub-cellular localization of caveolins and cavins, and baseline mitochondrial function and expression/phosphorylation of AKT or GSK3β. Thus, while dietary ALA might present a useful strategy for improving or restoring cardioprotection in T2D, the mechanistic basis of this benefit requires further investigation. Shifts in inflammatory and VEGF signaling identified here await further study, while modification of caveolar structure and biophysical properties remains a possible determinant.

## Figures and Tables

**Figure 1 nutrients-12-02679-f001:**
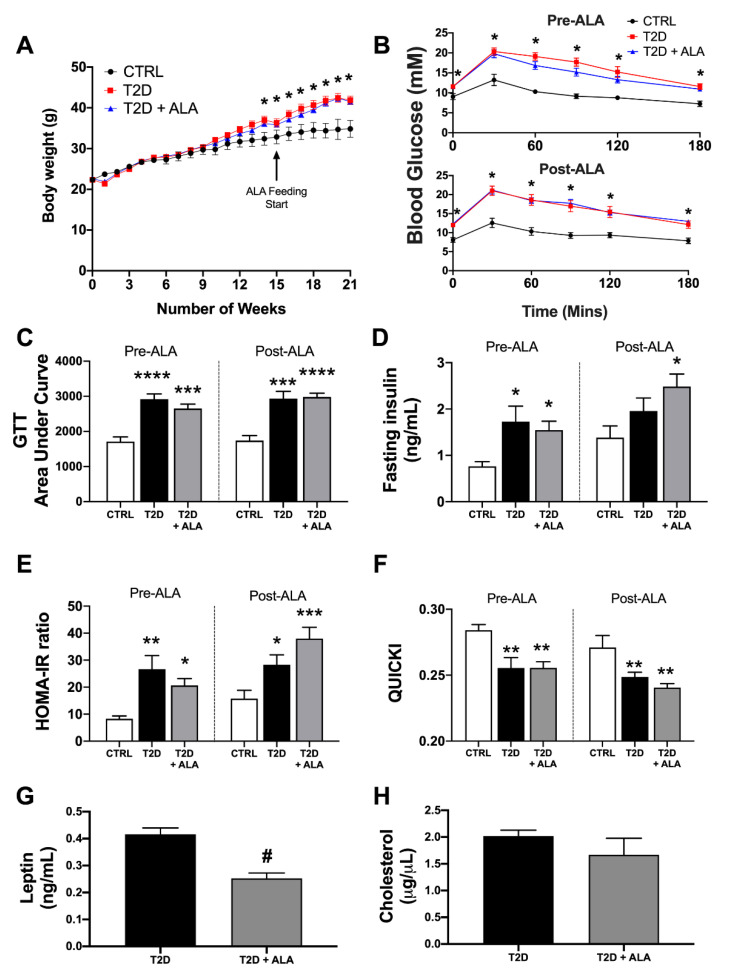
Metabolic phenotype in T2D mice ± dietary α-linolenic acid (ALA) supplementation. Data are shown for (**A**) body weight change over time (control [CTRL], *n* = 8; T2D, *n* = 21; T2D + ALA, *n* = 22); (**B**) glucose tolerance test (GTT) curve (CTRL, *n* = 8; T2D, *n* = 11; T2D + ALA, *n* = 12); (**C**) GTT area under curve (AUC) (CTRL, *n* = 8; T2D, *n* = 11; T2D + ALA, *n* = 12); (**D**) fasting serum insulin (CTRL, *n* = 8; T2D, *n* = 9; T2D + ALA, *n* = 10); (**E**) insulin resistance, based on homeostatic model assessment of insulin resistance (HOMA-IR) (CTRL, *n* = 8; T2D, *n* = 9; T2D + ALA, *n* = 10); (**F**) insulin sensitivity, based on quantitative insulin sensitivity check index (QUICKI) (CTRL, *n* = 8; T2D, *n* = 9; T2D + ALA, *n* = 10); (**G**) serum leptin (*n* = 8/group); and (**H**) total serum cholesterol (*n* = 7/group). Data in panels C–F were acquired at 14 weeks of western diet (1 week pre-ALA) and after an additional 6 weeks of continued feeding or dietary intervention (5 weeks post-ALA). Results presented as means ± SEM. *, *p* < 0.05 vs. CTRL; **, *p* < 0.01 vs. CTRL; ***, *p* < 0.001 vs. CTRL; ****, *p* < 0.0001 vs. CTRL; and #, *p* < 0.001 vs. T2D.

**Figure 2 nutrients-12-02679-f002:**
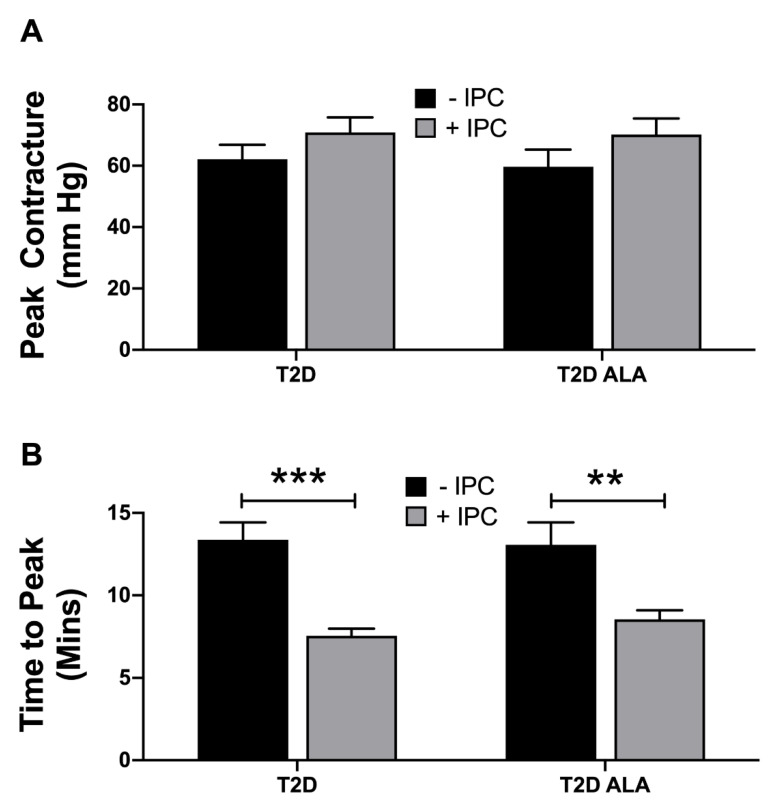
Ischemic contracture in untreated (−IPC) and preconditioned (+IPC) hearts from T2D mice ± dietary ALA supplementation. Data are shown for (**A**) peak contracture during 25 min ischemia; and (**B**) time to reach peak contracture. Results presented as means ± SEM (T2D (−IPC), *n* = 8; T2D (+IPC), *n* = 7; T2D + ALA (−IPC), *n* = 8; T2D + ALA (+IPC), *n* = 8). **, *p* < 0.01; and ***, *p* < 0.001.

**Figure 3 nutrients-12-02679-f003:**
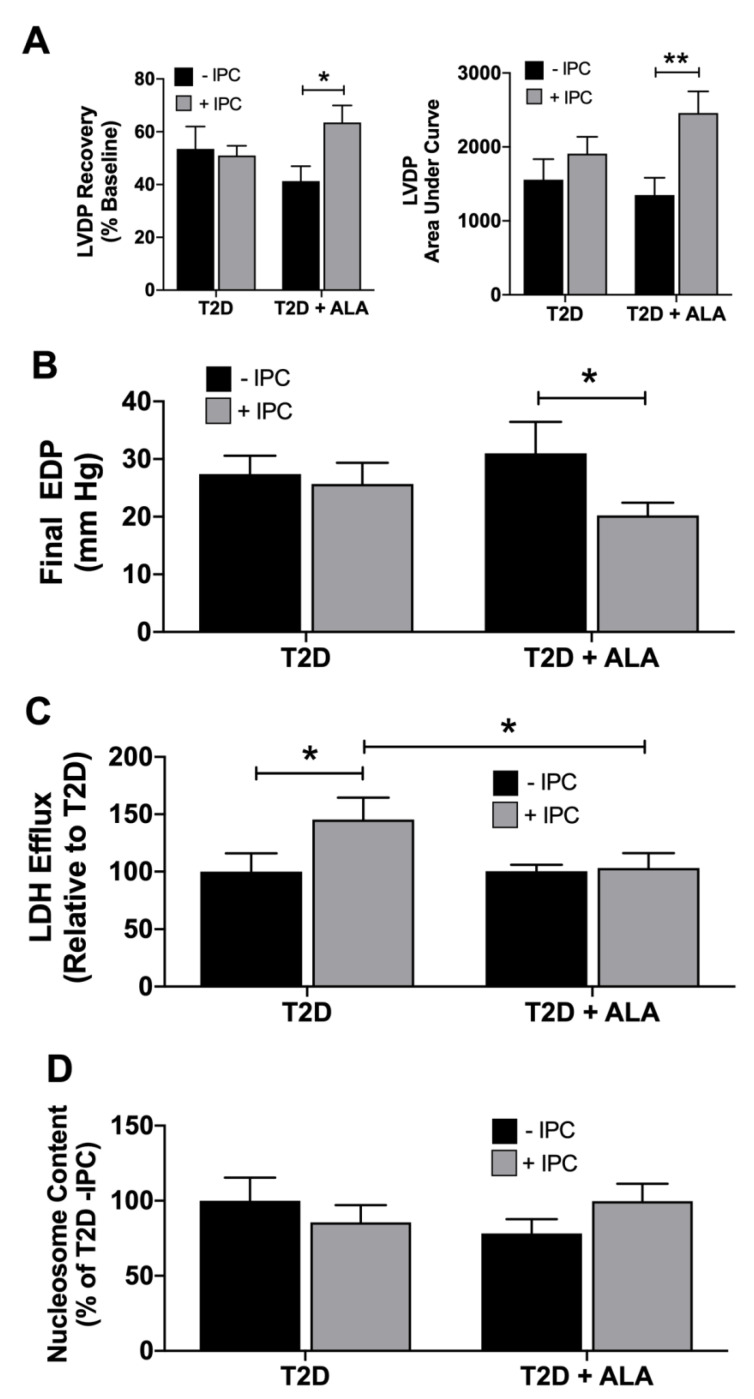
Post-ischemic outcomes in untreated (−IPC) and preconditioned (+IPC) hearts from T2D mice ± dietary ALA supplementation. Hearts were subjected to 25 min ischemia/45 min reperfusion with or without prior IPC (3 × 5 min ischemia/reperfusion). Data are shown for (**A**) final left ventricular developed pressure (LVDP), and LVDP area under the curve throughout reperfusion; (**B**) final end diastolic pressure (EDP); (**C**) total post-ischemic efflux of LDH; and (**D**) post-ischemic nucleosome content. Results presented as means ± SEM (T2D (−IPC), *n* = 8; T2D (+IPC), *n* = 7; T2D + ALA (−IPC), *n* = 8; T2D + ALA (+IPC), *n* = 8). *, *p* < 0.05; and **, *p* < 0.01.

**Figure 4 nutrients-12-02679-f004:**
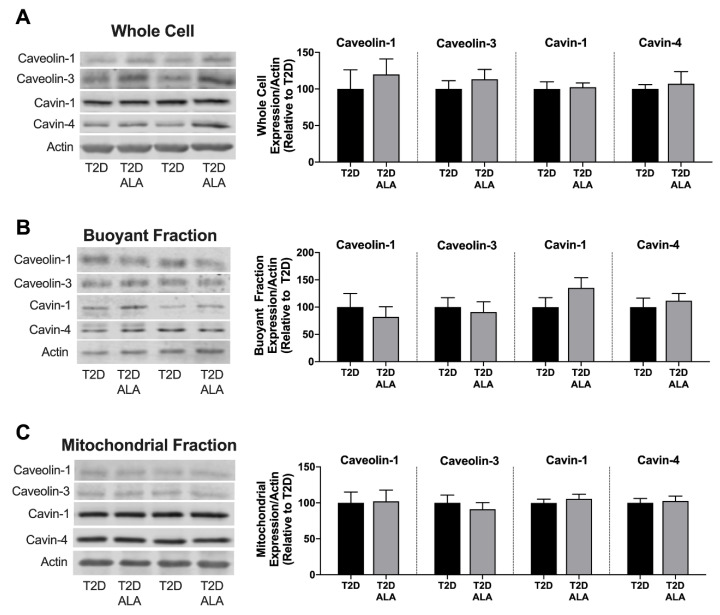
Total, buoyant membrane fraction and mitochondrial levels of caveolin and cavin proteins in hearts of T2D mice ± dietary ALA supplementation. Expression of caveolin-1, caveolin-3, cavin-1, and cavin-4 in (**A**) whole cell lysate; (**B**) buoyant (caveolae-enriched) fraction; and (**C**) mitochondrial fraction. Protein expression was normalized to actin in each blot and expressed relative to the untreated T2D group. Results presented as means ± SEM (*n* = 6/group for all proteins). No significant effects of ALA supplementation were detected.

**Figure 5 nutrients-12-02679-f005:**
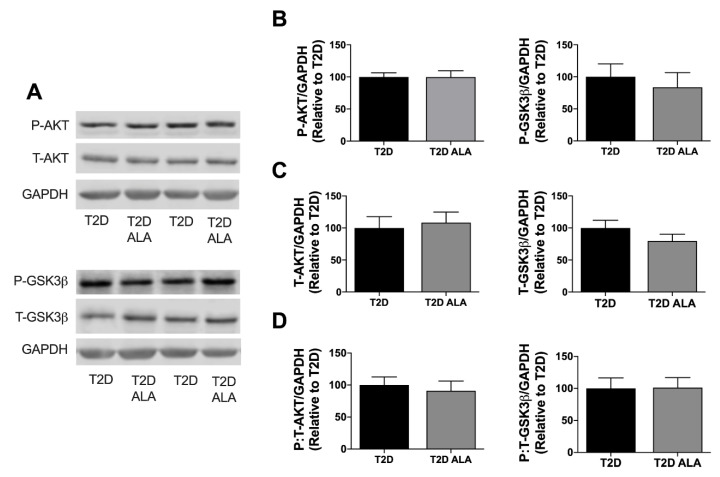
Cytosolic AKT and GSK3β expression and phosphorylation in myocardial lysates from T2D mice ± dietary ALA supplementation. (**A**) Representative blots; (**B**) phosphorylated AKT and GSK3β expression; (**C**) total AKT and GSK3β expression; (**D**) and ratio of phosphorylated:total AKT and GSK3β expression. Protein expression was normalized to GAPDH and expressed relative to the untreated T2D group. Results presented as means ± SEM (*n* = 6/group for all proteins). No significant effects of ALA supplementation were detected.

**Figure 6 nutrients-12-02679-f006:**
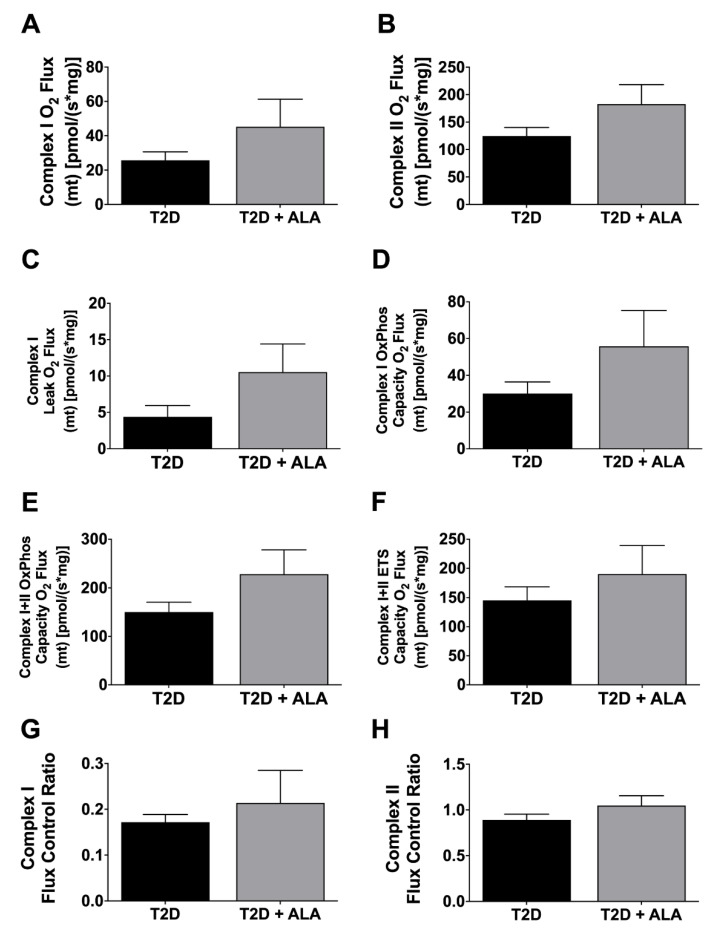
Mitochondrial respiratory function in ventricular myocardium from T2D mice ± dietary ALA supplementation. Data shown for (**A**) Complex I O_2_ Flux; (**B**) Complex II O_2_ Flux; (**C**) Complex I Leak O_2_ Flux; (**D**) Complex I oxidative phosphorylation (OxPhos) Capacity O_2_ Flux; (**E**) Complex I + II OxPhos Capacity O_2_ Flux; (**F**) Complex I + II electron transfer system (ETS) Capacity O_2_ Flux; (**G**) Complex I Flux Control Ratio; and (**H**) Complex II Flux Control Ratio. Results shown as means±SEM (*n* = 6/group). No significant effects of ALA supplementation were detected.

**Figure 7 nutrients-12-02679-f007:**
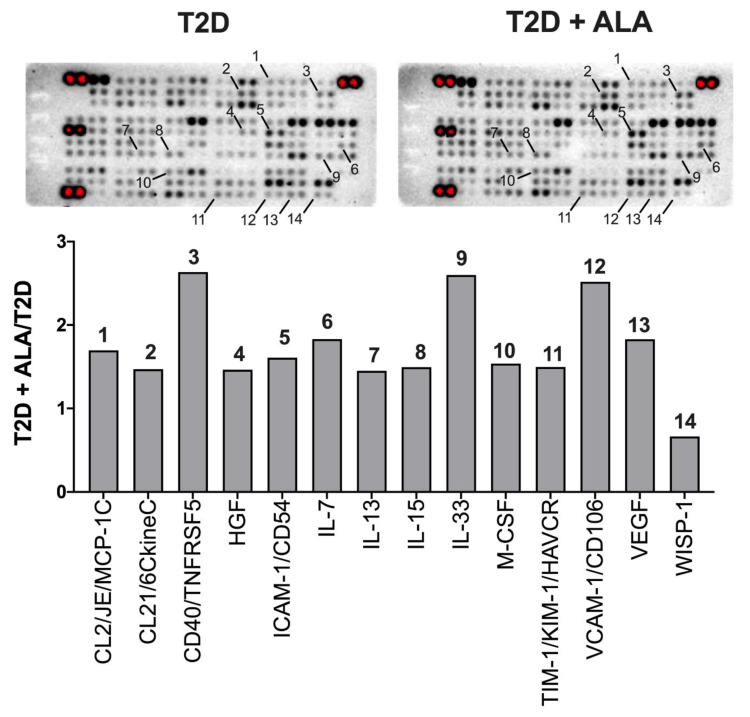
Inflammatory protein expression profiles in ventricular myocardium from T2D mice ± dietary ALA supplementation. Data are shown for analytes with ± 50% change in relative expression in left ventricles from ALA-supplemented T2D mice, compared to un-supplemented T2D mice. Analytes on membranes (in duplicate; upper panel) are labeled with numbers corresponding to bars on graph (lower panel). Results presented as a ratio (T2D ALA:T2D).

**Figure 8 nutrients-12-02679-f008:**
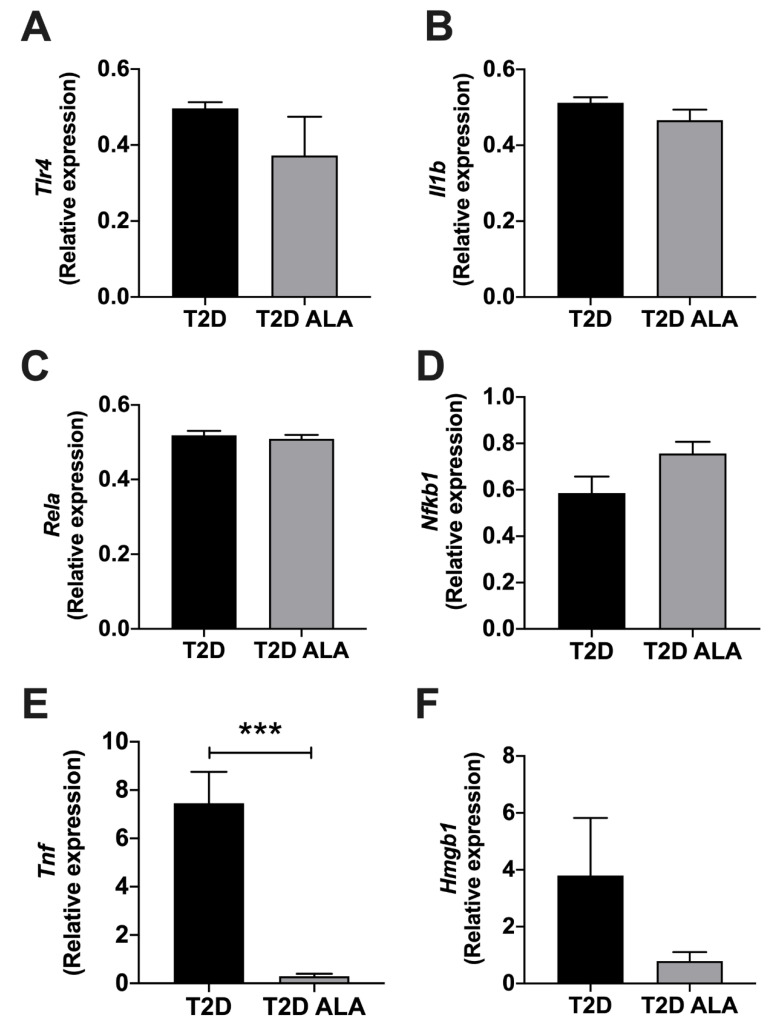
Inflammation related transcript expression in ventricular myocardium from T2D mice ± dietary ALA supplementation. Data are shown for gene expression as measured by real-time PCR in left ventricular tissue for (**A**) *Tlr4*, (**B**) *Il1b*, (**C**) *Rela*, (**D**) *Nfkb1*, (**E**) *Tnf*, and (**F**) *Hmgb1* from ALA-supplemented compared to un-supplemented T2D mice. ΔΔC_T_ values have been converted to 2^−ΔΔCT^ and presented as means ± SEM (*n* = 6/group). ***, *p* < 0.001.

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
