# Peer review of "Dietary α-Linolenic Acid Counters Cardioprotective Dysfunction in Diabetic Mice: Unconventional PUFA Protection"

_nutrients, 2020, doi:10.3390/nu12092679_

Round 1
Reviewer 1 Report
The work presented by Jake S Russell and colleagues showed that a novel impact of a-linolenic acid (ALA) on cardioprotective dysfunction in T2D mice, unrelated to caveolins/cavins, mitochondrial or stress kinase modulation. This study is novel, and the manuscript is well written. However, I just have some minor concerns.
- In the introduction section, it will be better to discuss a little bit about the biochemical function of ALA inside the cell.
- Please provide references in line 42 and 53.
- Extra space in line 67 between “mitochondrial membranes” and “has been reported”.
- The expression of some proteins varies between two groups of mice. For example, in Figure 4B, the expression of cavin-4 protein in T2D group is higher than in T2D+ALA group (left side), however, it is opposite in the second set of mice (right side). Please provide the images that represent the protein quantification.
- Please give the images of the western blots on the left side and then their quantification on the right side. Also, these are cropped very thin. if you have any wider crop images please provide that.
- We still don’t know whether the effect ALA only specific to the streptozotocin-induced T2D group. If you have any data on the effect of ALA on control nondiabetic mice, it is interesting to know that that too.
Author Response
- In the introduction section, it will be better to discuss a little bit about the biochemical function of ALA inside the cell.
We have now briefly discussed biochemistry of ALA and LA (pg 1-2, lines 43-52).
- Please provide references in line 42 and 53.
References are now included (pg 1, line 42 and pg 2, lines 63-64).
- Extra space in line 67 between “mitochondrial membranes” and “has been reported”.
The extra space has been deleted.
- The expression of some proteins varies between two groups of mice. For example, in Figure 4B, the expression of cavin-4 protein in T2D group is higher than in T2D+ALA group (left side), however, it is opposite in the second set of mice (right side). Please provide the images that represent the protein quantification.
Thank you for noticing this – the correct representative blots have now been included
- Please give the images of the western blots on the left side and then their quantification on the right side. Also, these are cropped very thin. if you have any wider crop images please provide that.
Representative blots are now on the left hand side in Fig 4 and 5 and we have cropped wider images.
- We still don’t know whether the effect ALA only specific to the streptozotocin-induced T2D group. If you have any data on the effect of ALA on control nondiabetic mice, it is interesting to know that that too.
As now clarified, this study was specifically designed to test for potential effects of ALA in a model of metabolic disease (T2D), testing for potential therapeutic benefits. It is not feasible to completely repeat the study and analyses to assess effects in healthy animals. Moreover, it is most relevant to assess benefits in disease rather than a state of health where intervention is unnecessary. We include phenotype data for control (healthy) mice for comparison to the disease model, and we refer to literature reporting effects of ALA in healthy tissue (ref 6-13, pg 1, line 41).
Reviewer 2 Report
Introduction
- Some of the abbreviations are not defined.
- What is the symbol at the end of line 74 (I presume a typo)?
Methods
- Why did you chose that specific dose of STZ? A look through the literature shows that others use a higher single dose or a multiple doses (similar to that used here) over a number of consecutive days. Did you check the degree of beta-cell destruction achieved with the dose that you used?
- How were group sizes chosen? They seem to vary a lot- for example, why are there 22 animals included in the body weight data in figure 1 but much smaller groups were included in the rest of the metabolic analysis (12 for GTT/ 8 for serum leptin etc.). This should be explained as this makes me worry that there is data missing. How did you decide which animals to include in which tests?
- What method did you use to analyse the qPCR data (e.g. delta delta Ct/ Pfaffl method etc.)?
Results
- I think figure 1 would benefit from inclusion of the control animal data in the graphs. For example, this would allow one to see easily that the STZ + HFD regime induces glucose intolerance/ insulin resistance (and would confirm that the STZ+ HFD regime is indeed inducing diabetes). I find appendix B confusing and I'm not really sure what comparisons you are trying to make with that data.
- I think the body weight data in Figure 1 would be better as an X/Y graph showing body weight progression across the whole study.
- Why have you not shown the curves for the GTTs in figure 1? Again, this would help with the interpretation of the data.
- The figure legend for figure 1 does not match up with the figure (where is the "relative weight change" data? Also, no explanation of the statistical significance of *** on figure 1G (this suggests that a significant effect of ALA has been found!).
- Actin is not evenly loaded in Figure 4B, so not sure how appropriate it is as a loading control here.
- The contrast is also way too high on all of the blots.
- Figure 4 has sections A-D which are not explained in the figure legend.
- Figure 6 figure legend explains statistical significance (** or **) but none of the data is significant (have the symbols been missed off the graphs?).
- Figure 8 has sections A-F which are not explained in the figure legend. There is also a statistical significant description explained in the legend that does not appear in the figure (**).
- No explanation of what the * mean in appendix B.
Author Response
- Some of the abbreviations are not defined.
We have now defined all abbreviations, where appropriate
- What is the symbol at the end of line 74 (I presume a typo)?
The symbol was a typo (was supposed to be ‘β’), and has been corrected.
Methods
- Why did you chose that specific dose of STZ? A look through the literature shows that others use a higher single dose or a multiple doses (similar to that used here) over a number of consecutive days. Did you check the degree of beta-cell destruction achieved with the dose that you used?
We have now better referenced and described the model (pg 3, lines 108-111). To clarify, the high or multiple STZ dosages the reviewer refers to are specifically employed to model severe hyperglycemia/uncontrolled T1D, with these high and prolonged levels destroying b-cells and resulting in severe hyperglycemia and hypo-insulinemia. In contrast, in modelling T2D, a single (vs. repeated) and moderate dose of STZ results in pancreatic stress without ablating the cells. Combined with prolonged high fat/high carbohydrate feeding, this models T2D (see refs 44,63-66). The resultant phenotype clearly confirms the predicted T2D outcomes, with maintained or elevated insulin (opposite to the hypoinsulinemia with high/prolonged STZ), combined with insulin resistance and hyperglycemia (Fig. 1D, E, F). This non-genetic model of chronic T2D development (β-cell stress + HFHC feeding) offers advantages over genetic models, and has been employed by other groups (see refs 44, 63-65)
- How were group sizes chosen? They seem to vary a lot- for example, why are there 22 animals included in the body weight data in figure 1 but much smaller groups were included in the rest of the metabolic analysis (12 for GTT/ 8 for serum leptin etc.). This should be explained as this makes me worry that there is data missing. How did you decide which animals to include in which tests?
We apologise for this confusion around the data in Figure 1, and clarify that body weight was measured in all 22 T2D mice, 22 T2D+ALA mice and 8 control mice. Of the 22 mice in each T2D group, 16 were sacrificed for cardiac perfusions (n=8 for IR; n=8 for IR+IPC) and the remaining n=6/group were sacrificed for cardiac mitochondrial respiration and protein/gene analyses in fresh tissue. This is described on pg 3, lines 119-122. While body weight was recorded for all mice, other metabolic measurements in Figure 1 were made on randomly selected subsets from T2D and T2D+ALA groups (n=13 mice per group). Of these, 2 mice were excluded as outliers according to a Grubbs test (pg 6, lines 282-283). Due to limited serum availability, leptin and cholesterol were measured in n=8 and n=7 mice/group], respectively. For controls, phenotype data was acquired in n=8 age-matched non-diabetic mice.
- What method did you use to analyse the qPCR data (e.g. delta delta Ct/ Pfaffl method etc.)?
Analysis of PCR data was conducted by employing the Delta-Delta Ct method (pg 6, line 277).
Results
- I think figure 1 would benefit from inclusion of the control animal data in the graphs. For example, this would allow one to see easily that the STZ + HFD regime induces glucose intolerance/ insulin resistance (and would confirm that the STZ+ HFD regime is indeed inducing diabetes). I find appendix B confusing and I'm not really sure what comparisons you are trying to make with that data.
We agree and have revised the Fig 1 to now include the control data (better highlighting the T2D phenotype).
- I think the body weight data in Figure 1 would be better as an X/Y graph showing body weight progression across the whole study.
We have replaced Fig 1A and B with an X/Y graph, now shown in Fig 1A.
- Why have you not shown the curves for the GTTs in figure 1? Again, this would help with the interpretation of the data.
GTT curves are now included in Fig 1B.
- The figure legend for figure 1 does not match up with the figure (where is the "relative weight change" data? Also, no explanation of the statistical significance of *** on figure 1G (this suggests that a significant effect of ALA hasbeen found!).
Fig 1 caption has now been corrected
- Actin is not evenly loaded in Figure 4B, so not sure how appropriate it is as a loading control here.
Thank you for noticing this – the correct representative blots have now been included
- The contrast is also way too high on all of the blots.
We have improved image quality/reduced contrast for representative blots in Fig. 4 and 5
- Figure 4 has sections A-D which are not explained in the figure legend.
Fig 4 caption has now been corrected
- Figure 6 figure legend explains statistical significance (** or **)but none of the data is significant (have the symbols been missed off the graphs?).
Fig 6 caption has now been corrected
- Figure 8 has sections A-F which are not explained in the figure legend. There is also a statistical significant description explained in the legend that does not appear in the figure (**).
Fig 8 caption has now been corrected
- No explanation of what the * mean in appendix B.
We have now removed Appendix B
Round 2
Reviewer 2 Report
Thank you for addressing all of my queries. However, I do still have a couple of points that I think need action:
I am still not clear as to the rationale of only performing the assessments of glucose handling in a subset of the animals and not in them all. Were there specific reasons for this? If so, these could be added to the methods section.
Figure 1 is much clearer with control data included and thank you for clarifying the group numbers. However, could you also please state which groups the "outlier" mice were excluded from? Fig 1B and C has n= 8 controls, 12 T2D and 12 T2D+ALA (so I would presume 1 outlier has been removed from each treatment group), however, group numbers are n = 8 vs 11 vs 13 for the fasting insulin, HOMA-IR and QUICKI (which would suggest based on this that the 2 outliers have been removed only from the T2D group). This is inconsistent- I think you should either completely remove or completely include the data from those animals in the whole analysis. Also, if the animals are outliers, is it really valid to them use them for the cardiac perfusion/ respiration and gene analysis? Could you please state which further analysis was performed in these outlier animals?
Author Response
Reviewer 2:
I am still not clear as to the rationale of only performing the assessments of glucose handling in a subset of the animals and not in them all. Were there specific reasons for this? If so, these could be added to the methods section.
Figure 1 is much clearer with control data included and thank you for clarifying the group numbers. However, could you also please state which groups the "outlier" mice were excluded from? Fig 1B and C has n= 8 controls, 12 T2D and 12 T2D+ALA (so I would presume 1 outlier has been removed from each treatment group), however, group numbers are n = 8 vs 11 vs 13 for the fasting insulin, HOMA-IR and QUICKI (which would suggest based on this that the 2 outliers have been removed only from the T2D group). This is inconsistent- I think you should either completely remove or completely include the data from those animals in the whole analysis. Also, if the animals are outliers, is it really valid to them use them for the cardiac perfusion/ respiration and gene analysis? Could you please state which further analysis was performed in these outlier animals?
Author Response:
We again apologise for confusion surrounding sample sizes. We now provide a schematic outlining the study design and n-values/sampling for the different analyses, better orientating the reader (new Appendix A).
To clarify, a total of 44 mice were subjected to the T2D (STZ/HFHC diet) protocol, with half receiving ALA in the final weeks (n=22/group). However, we note that one T2D mouse was identified as an outlier (not 2, as indicated previously - we apologise for this error), giving an n=21 for T2D and n=22 for T2D+ALA. This reduced the 'n' value for perfused T2D hearts + IPC, as follows: T2D, n=8; T2D+IPC, n=7; T2D +ALA, n=8; T2D + ALA + IPC, n=8.
As clarified in the revised Methods (lines 126-127) and Appendix A, body weights and fasting glucose were measured in all mice (final n=21 for T2D after outlier removal; n=22 for T2D + ALA). We now include complete fasting glucose datasets in a new Appendix B.
As now noted in revised Methods (lines 126-134) and Appendix A, to acquire representative metabolic data for T2D groups, while limiting significant costs involved, we undertook specific analyses in sub-sets of the T2D and T2D+ALA groups, as follows:
- GTTs were performed in 12 randomly selected mice from each group, giving n=11 for T2D (after outlier removal), and n=12 for T2D + ALA.
- Fasting insulin, HOMA and QUICKI were determined in 10 mice per group, giving n=9 for T2D (after outlier removal), and n=10 for T2D + ALA.
- Sufficient serum remained for analysis of leptin in n=8 mice/group, and cholesterol in n=7 mice/group.
Age-matched control comparator mice (n=8/group) were assessed to confirm a T2D phenotype in the experimental groups, with GTT, fasting glucose and insulin (and insulin resistance) analyses undertaken in all 8 mice.
We have now updated Figs 1, 2 and 3, and re-analysed all data accordingly. For transparency, we have also added further detail in the Methods section regarding group sizes (lines 126-134) and a descriptive outline of the study and 'n' values in Appendix A. Please note that the overall data and trends have not been affected in the new analysis. We hope that this clarifies group sizes in the study.